# Association between CYP17A1 rs743572 polymorphism and cancer risk: A meta-analysis

**Bin Wang**[1], **Zhumin Cao**[2]*, **Ying Li**[3]*, **Hua Zou**[4]*

**1** Department of Oncology, the Seventh People's Hospital of Chongqing (Affiliated Central Hospital of Chongqing University of Technology), Chongqing, China, **2** Vascular interventional department, the Seventh People's Hospital of Chongqing (Affiliated Central Hospital of Chongqing University of Technology), Chongqing, China, **3** Department of Pharmacy, Chongqing Seventh People's Hospital(Affiliated Central Hospital of Chongqing University of Technology), Chongqing, China, **4** Cancer Center, Daping Hospital, Army military Medical University, Chongqing, China

* 15922949835@163.com (ZC); liying200609@163.com (YL); zouhuadpyy@163.com (HZ)

## Abstract

The role of the CYP17A1 gene's rs743572 polymorphism in cancer susceptibility has been a subject of extensive investigation, yet existing evidence remains inconclusive. In this meta-analysis, we systematically reviewed and synthesized data from 29 studies to assess the CYP17A1 rs743572 polymorphism's relationship with cancer susceptibility. We strictly searched on EMBASE, PubMed, and Web of Science databases and explored rs743572 polymorphism's association with cancer risks according to search strategy, enrolling 29 studies (13,767 cases and 17,441 controls). rs743572 was markedly related to enhanced cancer susceptibility risk; FPRP and TSA analyses were employed for confirmation. According to cancer type-based stratified analyses, rs743572 exhibited a notable association with bladder cancer, breast cancer, non-Hodgkin lymphoma, and hepatocellular cancer. In conclusion, systematic meta-analysis suggests a significant role for the rs743572 polymorphism in cancer pathogenesis, with particular prominence observed in bladder cancer, breast cancer, non-Hodgkin lymphoma, and hepatocellular cancer.

## 1. Introduction

Cancer remarkably affect public health worldwide. As per the GLOBOCAN 2020 report, the anticipated figures for 2020 indicate a staggering 19.3 million new cancer cases and 10.0 million deaths globally [1]. A multitude of environment- and gene-related factors contribute to the emergence of cancer, though the underlying mechanisms remain incompletely elucidated [2].

Cytochrome P450 enzymes are crucial for human health, affecting various endogenous and exogenous components' metabolism [3]. Among these enzymes, CYP17A1, belonging to the cytochrome P450 family, is particularly pivotal for human health and its association with cancer [4]. CYP17A1 is involved in steroidogenesis, a

**Data availability statement:** All relevant data are within the paper and its Supporting Information files.

**Funding:** This research was funded by grants from Chongqing Young and Middle-aged Medical High-end Talents (No. YXGD202405), Chongqing District and County Head Goose Talents, Chongqing Science and Technology and Health Joint Scientific Research Project on Traditional Chinese Medicine (No. 2024ZYYB036), and Chongqing Banan District Science and Technology and Health Joint Scientific Research Project on Traditional Chinese Medicine (No. BNWJ202300112). The funders had no role in study design, data collection and analysis, decision to publish, or preparation of the manuscript.

**Competing interests:** The authors have declared that no competing interests exist.

crucial process of steroid hormones biosynthesis [5]. Specifically, CYP17A1 catalyzes the 17α-hydroxylation of pregnenolone and progesterone, producing intermediates that serve as precursors for synthesizing androgens and estrogens [5]. Dysregulated steroid hormone production, often mediated by CYP17A1, closely participates in the pathogenesis of multiple cancers: prostate cancer, hormone-sensitive breast cancer, etc. [6]. In prostate cancer, for instance, CYP17A1 is central for androgen biosynthesis, contributing to producing testosterone and dihydrotestosterone, which drive the growth of prostate tumors [6]. As a result, CYP17A1 has become a target for therapeutic intervention in prostate cancer treatment. Understanding the intricate relationship between cytochrome P450 enzymes, particularly CYP17A1, and cancer provides valuable insights into the development of targeted therapies aimed at modulating steroid hormone levels. By unraveling the molecular mechanisms underlying these processes, researchers strive to devise innovative strategies to mitigate the impact of dysregulated CYP17A1 activity on cancer progression, contributing to advancements in cancer treatment and improving overall human health.

In recent times, genome-wide association studies pinpointed a notable relationship between CYP17A1 polymorphisms and steroid hormone levels [7]. Notably, the CYP17A1 rs743572 polymorphism has been implicated in multiple cancers, spanning organs such as breast, lung, prostate, liver, etc. However, existing findings exhibit inconsistencies, likely stemming from limited sample sizes. To provide a more comprehensive understanding of this association, a series of analyses were conducted.

## 2. Materials and methods

### 2.1. Research retrieval

EMBASE, PubMed, and Web of Science were retrieved, covering studies up to December 20, 2023, to identify relevant research investigating the association between the CYP17A1 rs743572 polymorphism and cancer risks, as outlined in the search strategy detailed in Table 1. Four authors independently executed the search and screened the results.

### 2.2. Inclusion and exclusion criteria

The studies included in our analysis adhered to the following inclusion criteria: (A) Human-based research; (B) Case-control or cohort study design; (C) Availability of valid data; (D) Investigation into CYP17A1 rs743572 polymorphism's relationship with cancer; (E) Controls meeting Hardy-Weinberg Equilibrium (HWE), with $P > 0.05$. $P > 0.05$ represents genetic equilibrium in the population [8]. Additionally, the exclusion criteria was: (A) Case-only/non-cancer subject-only research; (B) Repetitive research; (C) Assembly abstracts.

### 2.3. Data screening

Valid data were screened independently by two researchers, encompassing first author's name, publication date, origin country, ethnicity, cancer type, and control source, as well as cases and controls' numbers in each study.

**Table 1. The detailed search strategies of lCYP17A1 rs743572 polymorphism's association with cancer susceptibility.**

| Source | Search strategy | |
|--------|----------------|---|
| Pubmed | #1: Polymorphism, genetic<br>#2: Polymorphism*<br>#3: SNP<br>#4: Single nucleotide polymorphism<br>#5: Variant<br>#6: Mutation<br>#7: #1 OR #2 OR #3 OR #4 OR #5 OR #6<br>#8: rs743572<br>#9: CYP17A1<br>#10: #8 AND #9<br>#11: Neoplasms<br>#12: Cancer<br>#13: Carcino*<br>#14: #11 OR #12 OR #13<br>#15: #7 AND #10 AND #14 | ((Polymorphism, genetic) OR Polymorphism* OR SNP OR (Single nucleotide polymorphism) OR Variant OR Mutation) AND (rs743572 AND CYP17A1) AND (Neoplasms OR Cancer OR Carcino*)) |
| Embase | #1: 'neoplasm'/exp<br>#2: cancer<br>#3: tumor<br>#4: carcinoma<br>#5: carcinogenesis<br>#6: #1 OR #2 OR #3 OR #4 OR #5<br>#7: rs743572<br>#8: CYP17A1<br>#9: #7 AND #8<br>#10: 'single nucleotide polymorphism'/exp<br>#11: SNP<br>#12: polymorphism<br>#13: variant<br>#14: mutation<br>#15: #10 OR #11 OR #12 OR #13 OR #14<br>#16: #6 AND #9 AND #15 | (('neoplasm'/exp OR cancer OR tumor OR carcinoma OR carcinogenesis) AND (rs743572 AND CYP17A1) AND ('single nucleotide polymorphism'/exp OR SNP OR polymorphism OR variant OR mutation)) |

## 2.4. Quality evaluation

Each study's quality was independently evaluated by two researchers with the quality assessment criteria outlined in S1 Table, as previously described [9]. The following aspects were involved: cancer case ascertainment (0–2), case representation (0–2), control representation (0–3), control selection (0–2), genotyping examination (0–2), adherence to HWE (0–1), and total sample size (0–3). Total scores ranged from 0 to 15, with studies scoring > 9 points categorized as high quality.

## 2.5. Statistical analysis

Statistical analysis was conducted using Stata software (Version 12.0, StataCorp, College Station, TX, USA). The association between the CYP17A1 rs743572 polymorphism and cancer risks was assessed using odds ratio (OR) and 95% confidence interval (CI). Dominant, recessive, homozygote, heterozygote, and allele models were employed for analysis. Cochrane Q-test and P-values were applied for heterogeneity evaluation: random-effects model with $P \leq 0.10$ or $I^2 \geq 50\%$; fixed-effects model otherwise. Information was stratified according to ethnicity, control source, and cancer type. Publication bias was assessed through Begg's test and Egger's test. Study reliability was evaluated by analyzing sensitivity. $P < 0.05$ represented statistical significance. False positive report probability (FPRP) analysis was carried out as previously mentioned [10], which assumed 0.1 as the prior probability, with an FPRP threshold of 0.2 considered noteworthy [11]. To address potential biases from sparse data that may impact meta-analyses, trial sequential analysis (TSA) was conducted using the Copenhagen Trial Unit's software (Denmark, 2011). The study parameters included overall type–I error (5%), test power (80%), and relative risk reduction (20%).

## 3. Results

### 3.1. Study selection

Our search strategy yielded 428 articles. Following evaluating titles and abstracts, 35 articles met our inclusion criteria. Upon a thorough examination of the full texts, 8 articles were excluded. This exclusion comprised three articles that did not address the association between CYP17A1 rs743572 polymorphism and cancer susceptibility, two review articles, and three that lacked detailed genotyping data. Ultimately, 27 articles, encompassing 29 studies (13,767 cases; 17,441 controls) were included [12–38]. The screening process is illustrated in Fig 1.

Broadly, the meta-analysis involved 21 studies with Caucasian populations and 8 studies with Asian populations. Out of the total, 14 studies focused on the impact of CYP17A1 rs743572 polymorphism in breast cancer, 1 in bladder cancer, 2 in colorectal cancer, 2 in pancreatic cancer, 6 in prostate cancer, 1 in endometrial cancer, 1 in non-Hodgkin lymphoma, 1 in hepatocellular cancer, and 1 in non-small cell lung cancer (Table 2).

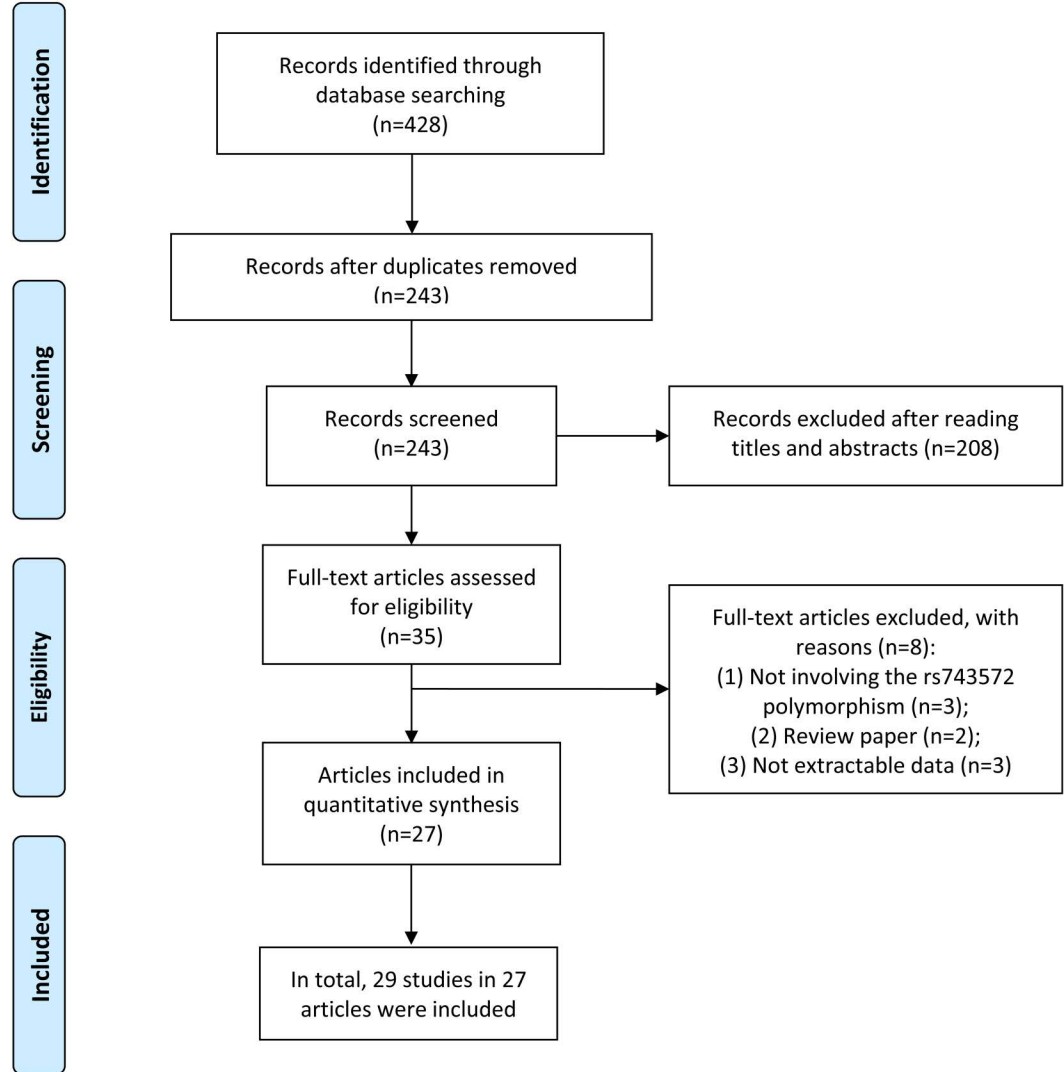

**Fig 1. Study identification and selection.**

## Table 2. Details of included studies.

| First Author | Year | Ethnicity | Cancer type | Control source | Geno-typing method | Cases (CC/TC/TT) | Controls (CC/TC/TT) | HWE | Score |
|---|---|---|---|---|---|---|---|---|---|
| Bethke | 2007 | Caucasian | Colorectal cancer | PB | Illumina | 995/1192/374 | 1045/1234/416 | 0.102 | 12 |
| Chattopadhyay | 2014 | Asian | Breast cancer | PB | PCR-RFLP | 230/116/14 | 260/93/7 | 0.692 | 10 |
| Duell | 2010 | Caucasian | Pancreatic cancer | PB | MassAR-RAY | 101/113/32 | 287/372/141 | 0.283 | 7 |
| Farzaneh | 2016 | Asian | Breast cancer | HB | PCR-RFLP | 32/70/22 | 27/56/17 | 0.189 | 6 |
| Ghisari | 2014 | Caucasian | Breast cancer | HB | TaqMan | 24/57/32 | 12/12/6 | 0.882 | 7 |
| Ghisari | 2017 | Caucasian | Breast cancer | HB | TaqMan | 52/65/23 | 71/88/33 | 0.524 | 10 |
| He | 2022 | Asian | Breast cancer | HB | MassAR-RAY | 119/131/52 | 183/89/30 | 0.128 | 9 |
| Hussain | 2016 | Asian | Pancreatic cancer | HB | PCR-RFLP | 72/122/61 | 119/154/47 | 0.805 | 7 |
| Iwasaki-1 | 2010 | Asian | Breast cancer | HB | TaqMan | 111/189/88 | 122/182/84 | 0.299 | 10 |
| Iwasaki-2 | 2010 | Caucasian | Breast cancer | HB | TaqMan | 17/48/13 | 23/33/23 | 0.144 | 10 |
| Iwasaki-3 | 2010 | Caucasian | Breast cancer | HB | TaqMan | 135/185/59 | 130/200/49 | 0.139 | 11 |
| Karakus | 2015 | Caucasian | Breast cancer | HB | PCR-RFLP | 102/79/18 | 104/78/15 | 0.143 | 8 |
| Martinez-Gonzalez | 2020 | Caucasian | Prostate cancer | HB | TaqMan | 89/67/5 | 95/61/6 | 0.317 | 7 |
| Olson | 2008 | Caucasian | Endometrial cancer | PB | PCR-RFLP | 122/207/71 | 148/210/59 | 0.297 | 9 |
| Martin-Way | 2022 | Caucasian | Bladder cancer | HB | TaqMan | 58/43/6 | 78/21/0 | 0.589 | 10 |
| MARIE-GENICA Consortium on Genetic Susceptibility for Meno-pausal Hormone Therapy Related Breast Cancer Risk | 2010 | Caucasian | Breast cancer | PB | MassAR-RAY | 1043/1573/529 | 1834/2712/941 | 0.254 | 12 |
| Reding | 2012 | Caucasian | Breast cancer | PB | PCR-RFLP | 241/356/50 | 234/315/100 | 0.722 | 13 |
| Risio | 2011 | Caucasian | Prostate cancer | PB | SNPlex assay | 142/62/28 | 45/20/4 | 0.388 | 11 |
| Rizzolo | 2019 | Caucasian | Breast cancer | HB | TaqMan | 190/301/106 | 347/489/186 | 0.550 | 11 |
| Robles-Fernandez | 2017 | Caucasian | Prostate cancer | HB | TaqMan | 67/66/23 | 53/82/20 | 0.178 | 9 |
| Rudolph | 2011 | Caucasian | Colorectal cancer | HB | PCR-RFLP | 213/355/115 | 252/326/103 | 0.885 | 11 |
| Sakoda | 2008 | Caucasian | Breast cancer | PB | PCR-RFLP | 216/297/102 | 298/441/138 | 0.232 | 7 |
| Skibola | 2005 | Caucasian | Non-Hodgkin lymphoma | PB | SNPlex assay | 237/274/93 | 308/366/83 | 0.095 | 14 |
| Song | 2016 | Asian | Prostate cancer | HB | PCR-RFLP | 38/76/62 | 43/71/54 | 0.050 | 10 |
| Tang | 2018 | Caucasian | Prostate cancer | HB | SNPlex assay | 219/323/73 | 189/259/70 | 0.205 | 7 |
| TÜZÜNER | 2010 | Caucasian | Breast cancer | HB | PCR-RFLP | 18/27/10 | 38/44/9 | 0.466 | 9 |
| Mattia | 2017 | Caucasian | Hepatocellular cancer | HB | TaqMan | 39/88/40 | 72/84/33 | 0.325 | 10 |
| Zhang | 2013 | Asian | Non-small cell lung cancer | HB | MassAR-RAY | 59/108/37 | 72/91/37 | 0.386 | 11 |
| Wu | 2020 | Asian | Prostate cancer | HB | TaqMan | 14/33/11 | 25/27/6 | 0.744 | 7 |

Note: PB: publication-based controls; HB: hospital-based controls; C: wild type; T: mutated type.

## 3.2. Meta-analysis of rs743572

As revealed by the results (Table 3), rs743572 polymorphism notably elevated cancer risk in dominant (OR=1.09, 95% CI = 1.04–1.14), heterozygote (OR=1.09, 95% CI = 1.04–1.15), homozygote (OR=1.08, 95% CI = 1.01–1.16), and allele (OR=1.05, 95% CI = 1.01–1.08, Fig 2) models. When studies were stratified by ethnicity, significant associations were found in Asians (dominant, OR=1.50, 95% CI = 1.30–1.72; recessive, OR=1.32, 95% CI = 1.11–1.58; homozygote, OR=1.60, 95% CI = 1.31–1.96, Fig 3; heterozygote, OR=1.43, 95% CI = 1.24–1.66; Allele, OR=1.32, 95% CI = 1.20–1.46) and Caucasian populations (heterozygote, OR=1.05, 95% CI = 1.00–1.11). When studies were stratified by cancer type, rs743572 polymorphism increased bladder cancer risk (dominant, OR=3.14, 95% CI = 1.70–5.80, Fig 4; heterozygote, OR=2.75, 95% CI = 1.48–5.13; Allele, OR=2.92, 95% CI = 1.69–5.04), breast cancer (dominant, OR=1.08, 95% CI = 1.01–1.15, Fig 4; heterozygote, OR=1.08, 95% CI = 1.01–1.16), non-Hodgkin lymphoma (recessive, OR=1.48, 95% CI = 1.08–2.03; homozygote, OR=1.46, 95% CI = 1.04–2.05), and hepatocellular cancer (dominant, OR=2.02, 95% CI = 1.27–3.21,

**Table 3. Meta-analysis of CYP17A1 rs743572 polymorphism's association with cancer risk.**

| Variables | No. of study | No. of cases/controls | Dominant model (TT+TC vs. CC) OR (95%CI)/I2% | Recessive model (TT vs. TC+CC) OR (95%CI)/I2% | Homozygote model (TT vs. CC) OR (95%CI)/I2% | Heterozygote model (TC vs. CC) OR (95%CI)/I2% | Allele model (T vs. C) OR (95%CI)/I2% |
|---|---|---|---|---|---|---|---|
| Overall | 29 | 13773/17661 | **1.09 (1.04, 1.14)/63.7** | 1.03 (0.97, 1.10)/56.8 | **1.08 (1.01, 1.16)/63.5** | **1.09 (1.04, 1.15)/53.6** | **1.05 (1.01, 1.08)/69.5** |
| Caucasian | 21 | 11910/15761 | 1.04 (0.99, 1.10)/50.9 | 0.99 (0.93, 1.06)/56.9 | 1.02 (0.95, 1.10)/58.2 | **1.05 (1.00, 1.11)/41.6** | 1.01 (0.98, 1.05)/57.6 |
| Asian | 8 | 1683/1900 | **1.50 (1.30, 1.72)/48.9** | **1.32 (1.11, 1.58)/27.1** | **1.60 (1.31, 1.96)/42.5** | **1.43 (1.24, 1.66)/34.9** | **1.32 (1.20, 1.46)/62.7** |
| Bladder cancer | 1 | 99/107 | **3.14 (1.70, 5.80)/-** | 12.74 (0.71, 229.24)/- | 17.44 (0.96, 315.86)/- | **2.75 (1.48, 5.13)/-** | **2.92 (1.69, 5.04)/-** |
| Colorectal cancer | 2 | 3244/3376 | 1.05 (0.95, 1.16)/76.3 | 0.98 (0.85, 1.12)/25.1 | 1.01 (0.87, 1.17)/69.7 | 1.06 (0.96, 1.18)/68.2 | 1.02 (0.95, 1.09)/74.9 |
| Breast cancer | 14 | 7059/10236 | **1.08 (1.01, 1.15)/64.3** | 0.99 (0.91, 1.07)/62.9 | 1.03 (0.94, 1.13)/65.7 | **1.08 (1.01, 1.16)/55.8** | 1.03 (0.99, 1.08)/71.2 |
| Pancreatic cancer | 2 | 563/1284 | 1.04 (0.83, 1.30)/86.1 | 1.10 (0.83, 1.46)/90.1 | 1.11 (0.81, 1.52)/92.3 | 1.02 (0.80, 1.30)/64.3 | 1.05 (0.90, 1.22)/92.3 |
| Prostate cancer | 6 | 1398/1130 | 1.07 (0.91, 1.27)/37.9 | 1.07 (0.85, 1.35)/0 | 1.11 (0.85, 1.44)/24.1 | 1.06 (0.89, 1.26)/34.7 | 1.05 (0.94, 1.19)/30.9 |
| Endometrial cancer | 1 | 417/400 | 1.25 (0.94, 1.68)/- | 1.31 (0.90, 1.91)/- | 1.46 (0.96, 2.22)/- | 1.20 (0.88, 1.63)/- | 1.00 (0.82, 1.22)/- |
| Non-Hodgkin lymphoma | 1 | 604/757 | 1.06 (0.85, 1.32)/- | **1.48 (1.08, 2.03)/-** | **1.46 (1.04, 2.05)/-** | 0.97 (0.77, 1.23)/- | 1.14 (0.97, 1.33)/- |
| Hepatocellular cancer | 1 | 189/167 | **2.02 (1.27, 3.21)/-** | 1.49 (0.89, 2.50)/- | **2.24 (1.22, 4.09)/-** | **1.93 (1.18, 3.16)/-** | **1.54 (1.14, 2.07)/-** |
| Non-small cell lung cancer | 1 | 200/204 | 1.38 (0.91, 2.10)/- | 0.98 (0.59, 1.62)/- | 1.22 (0.69, 2.16)/- | 1.45 (0.93, 2.26)/- | 1.15 (0.87, 1.52)/- |
| HB | 20 | 4884/5403 | **1.25 (1.15, 1.36)/61.9** | **1.15 (1.03, 1.28)/25.8** | **1.30 (1.15, 1.46)/43.1** | **1.23 (1.13, 1.35)/54.4** | **1.16 (1.09, 1.23)/64.7** |
| PB | 9 | 8889/12258 | 1.02 (0.96, 1.08)/27.4 | 0.97 (0.90, 1.05)/76.2 | 0.98 (0.90, 1.07)/73.7 | 1.02 (0.96, 1.09)/0 | 0.99 (0.95, 1.03)/59.8 |
| High quality (> 9) | 16 | 10478/13404 | **1.07 (1.02, 1.13)/55.2** | 0.99 (0.93, 1.07)/62.6 | 1.04 (0.96, 1.12)/61.6 | **1.08 (1.02, 1.14)/46.5** | 1.03 (0.99, 1.07)/65.3 |
| Low quality (≤ 9) | 13 | 3295/4257 | **1.15 (1.04, 1.26)/71.7** | **1.16 (1.01, 1.33)/42.0** | **1.23 (1.06, 1.42)/64.5** | **1.12 (1.01, 1.24)/62.4** | **1.09 (1.02, 1.17)/74.5** |

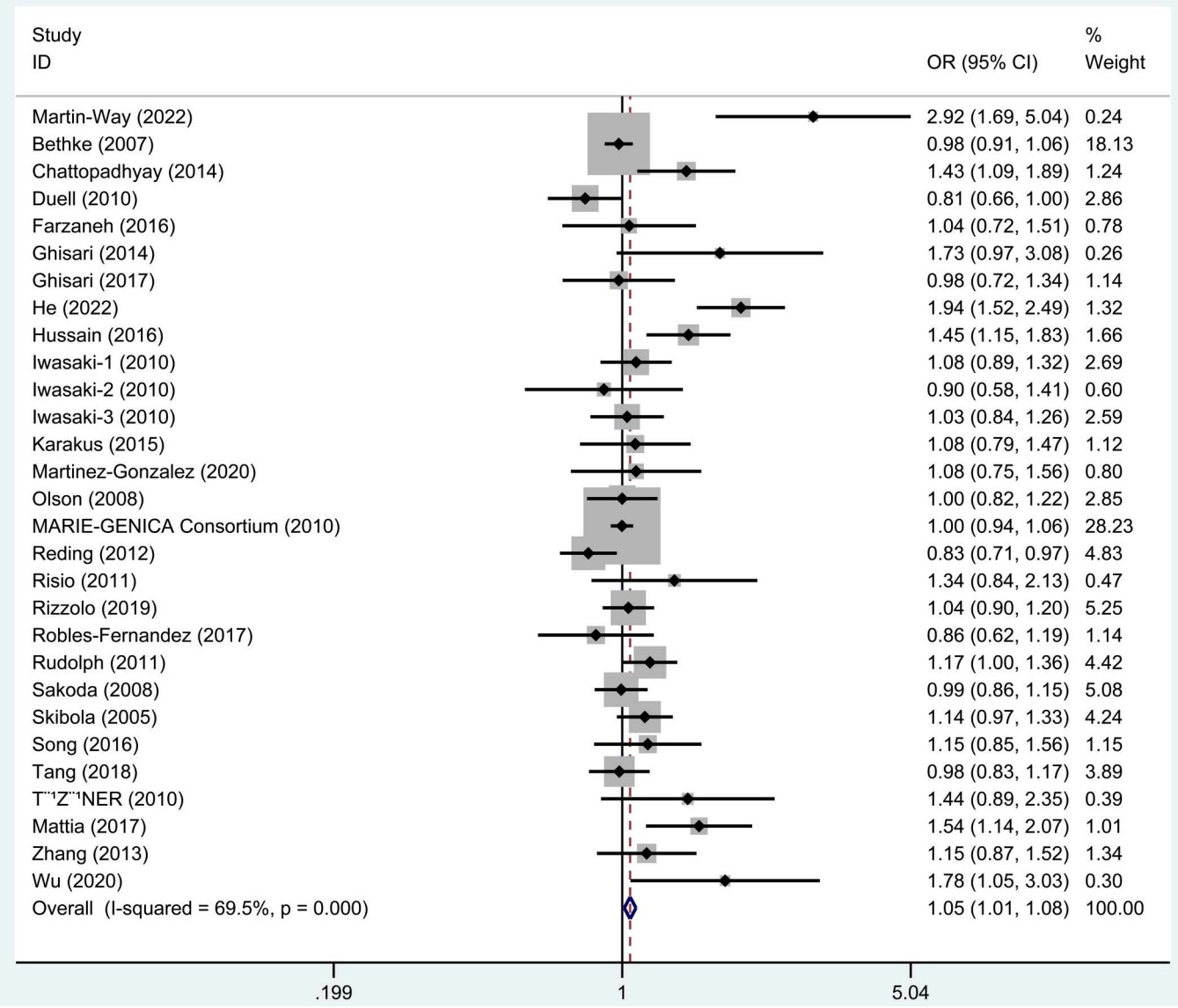

**Fig 2. Meta-analysis for rs743572 polymorphism's association with cancer risk (allele model).** Each study is represented by a square (point estimate) and a horizontal line (95% confidence interval [CI]). The size of the square corresponds to the study's weight in the meta-analysis. The diamond summarizes the overall pooled odds ratio (OR) and 95% CI. Significant associations (OR > 1) indicate increased cancer risk with the allele.

Fig 4; homozygote, OR=2.24, 95% CI = 1.22–4.09; heterozygote, OR=1.93, 95% CI = 1.18–3.16; Allele, OR=1.54, 95% CI = 1.14–2.07). Additionally, a remarkable association between rs743572 polymorphism and elevated cancer risk was noted in studies with publication-based controls and hospital-based controls (Fig 5). According to Table 4, With a prior probability assumed as 0.1, associations with statistical significance were noteworthy (FPRP< 0.2) for overall cancer risk (dominant, heterozygote, homozygote, and allele models), Asians (dominant, recessive, homozygote, heterozygote, and allele models), Caucasian populations (heterozygote model), bladder cancer (dominant, heterozygote, and allele models), breast cancer (dominant and heterozygote models), non-Hodgkin lymphoma (recessive and homozygote models), hepatocellular cancer (dominant, homozygote, heterozygote, and allele models) subgroups (Table 4).

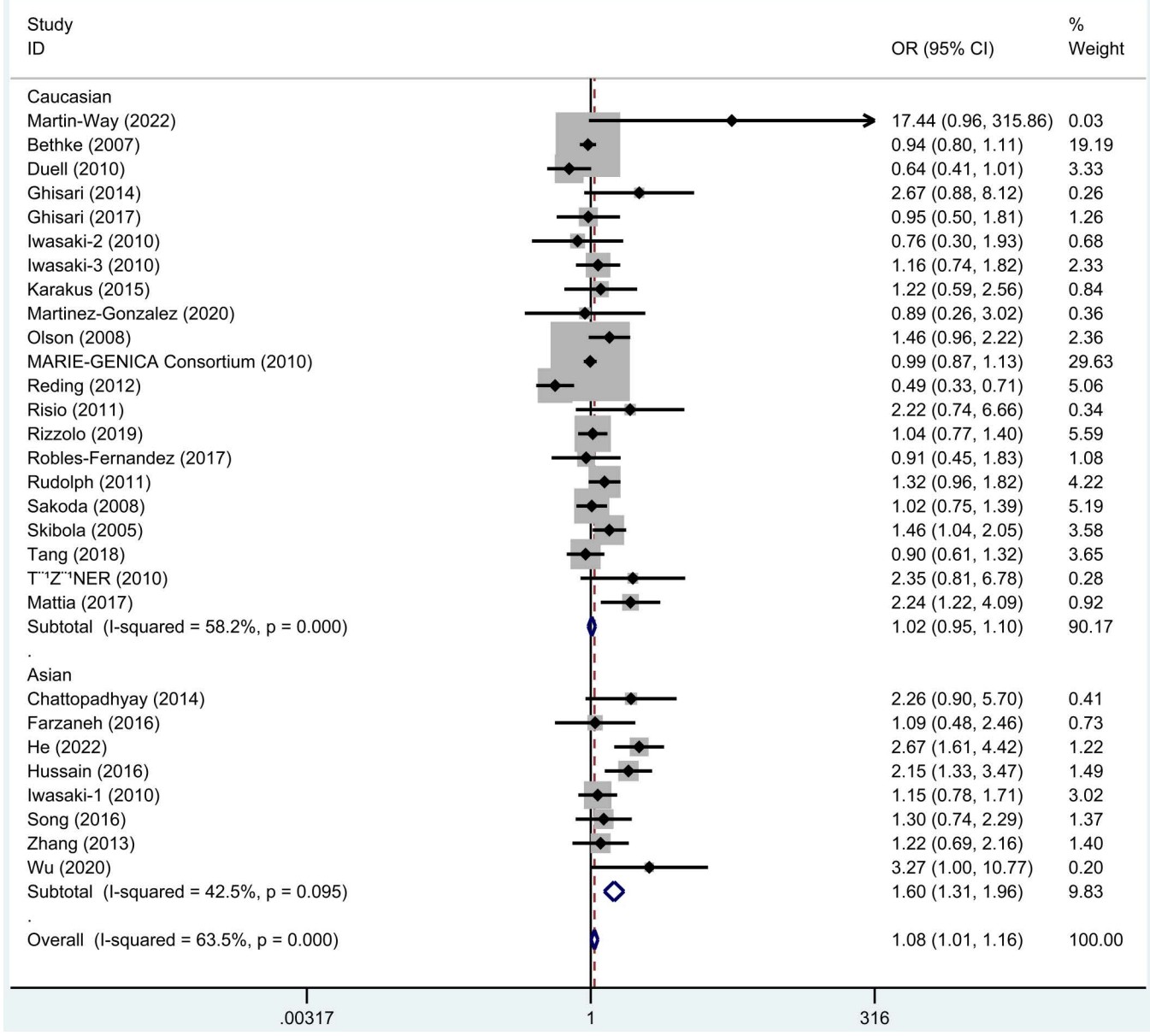

**Fig 3. Meta-analysis for rs743572 polymorphism's association with cancer risk stratified by ethnicity (homozygote model).** Subgroup analyses for Caucasian and Asian populations are shown. Significant associations were found in Asians (dominant, OR=1.50, 95% CI = 1.30-1.72; recessive, OR=1.32, 95% CI = 1.11-1.58; homozygote, OR=1.60, 95% CI = 1.31-1.96; heterozygote, OR=1.43, 95% CI = 1.24-1.66; Allele, OR=1.32, 95% CI = 1.20-1.46) and Caucasian populations (heterozygote, OR=1.05, 95% CI = 1.00-1.11).

### 3.3. TSA of rs743572

As depicted in Fig 6, even though case total number did not surpass the O'Brien-Fleming boundary, the cumulative Z-curve surpassed the test sequence monitoring boundary. This observation corroborates that rs743572 is markedly linked to cancer risk.

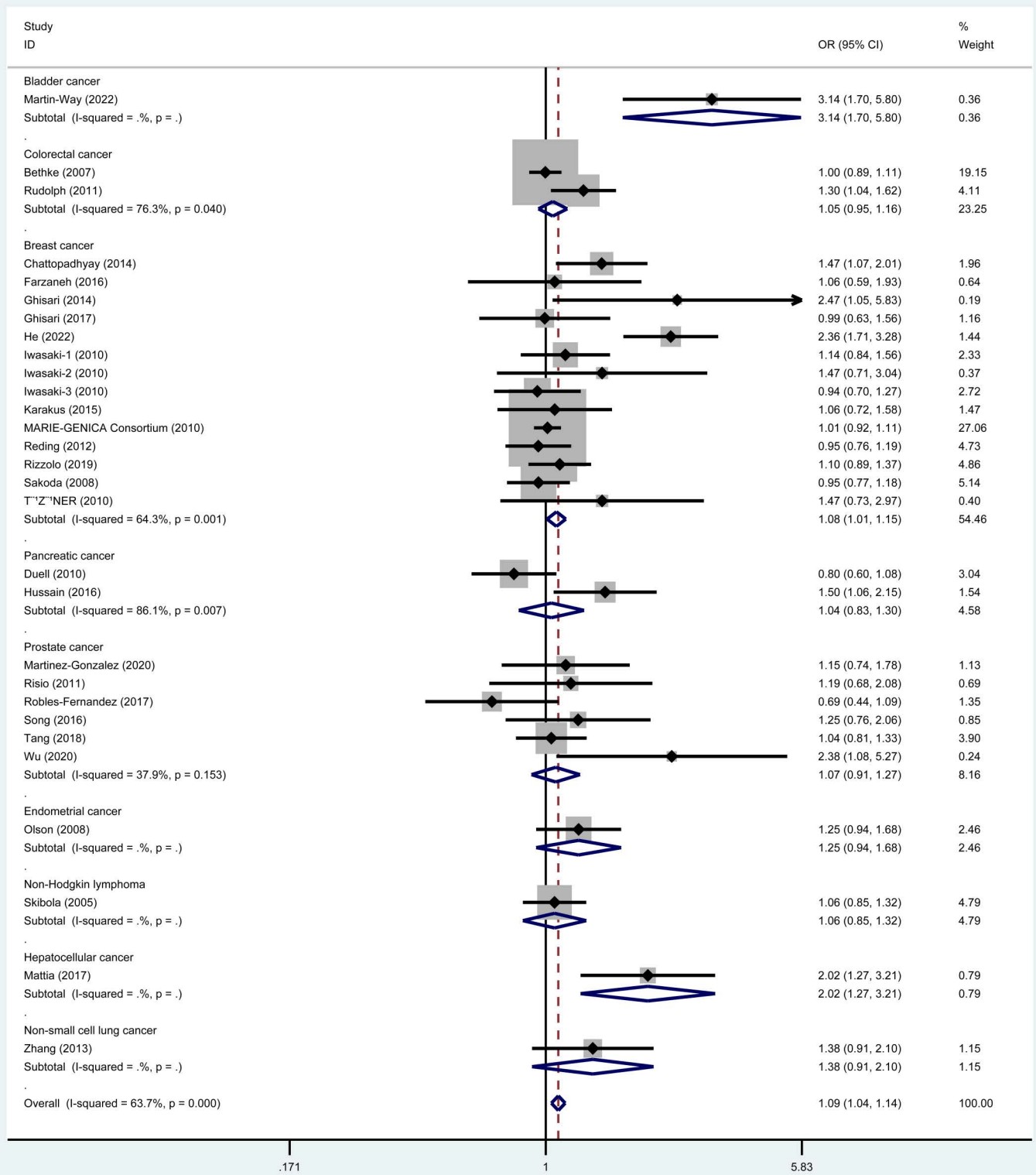

**Fig 4. Meta-analysis for rs743572 polymorphism's association with cancer risk stratified by cancer type (dominant model).** Stratification by cancer type revealed significant risk increases for bladder cancer (OR = 3.14, 95% CI = 1.70–5.80), breast cancer (OR = 1.08, 95% CI = 1.01–1.15), and hepatocellular carcinoma (OR = 2.02, 95% CI = 1.27–3.21).

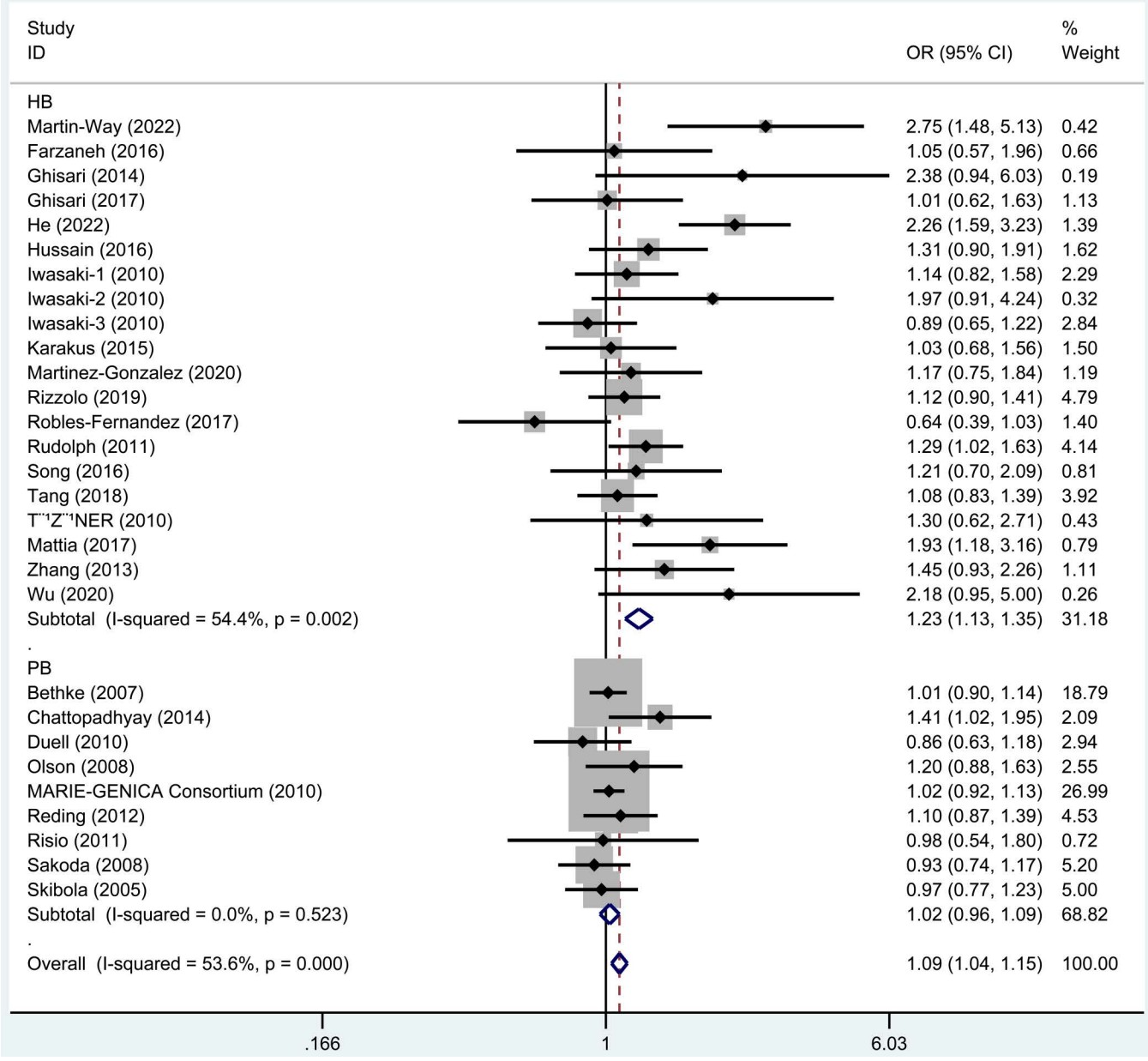

**Fig 5. Meta-analysis for rs743572 polymorphism's association with cancer risk stratified by control source (heterozygote model).** Hospital-based (HB) controls showed a significant association (OR = 1.23, 95% CI = 1.13–1.35; I²=54.4%), whereas population-based (PB) controls did not (OR = 1.02, 95% CI = 0.96–1.09; I²=0.0%). The overall effect remained significant (OR = 1.09, 95% CI = 1.04–1.15; I²=53.6%), suggesting potential selection bias in HB studies.

## 3.4. Publication bias

Both Begg's and Egger's tests revealed no substantial publication bias this meta-analysis (Figure 7A; Table 5). Moreover, as demonstrated by sensitivity analysis, no individual study notably altered the conclusion (Figure 7B).

**Table 4. FPRP values for CYP17A1 rs743572 polymorphism's association with cancer risk.**

| Significant association | OR (95%CI) | Prior probability | | | | |
|---|---|---|---|---|---|---|
| | | 0.25 | 0.1 | 0.01 | 0.001 | 0.0001 |
| **Dominant model (TT+TC vs. CC)** | | | | | | |
| Overall | 1.09 (1.04, 1.14) | **0.036** | **0.147** | 0.342 | 0.937 | 0.998 |
| Asian | 1.50 (1.30, 1.72) | **0.011** | **0.089** | **0.112** | **0.145** | 0.289 |
| Bladder cancer | 3.14 (1.70, 5.80) | **0.003** | **0.009** | **0.013** | **0.056** | **0.132** |
| Breast cancer | 1.08 (1.01, 1.15) | **0.046** | **0.153** | 0.267 | 0.463 | 0.895 |
| Hepatocellular cancer | 2.02 (1.27, 3.21) | **0.024** | **0.087** | **0.133** | **0.185** | 0.965 |
| HB | 1.25 (1.15, 1.36) | **0.005** | **0.012** | **0.114** | 0.565 | 0.927 |
| High quality (> 9) | 1.07 (1.02, 1.13) | **0.026** | **0.174** | 0.643 | 0.864 | 0.995 |
| Low quality (≤ 9) | 1.15 (1.04, 1.26) | **0.008** | **0.024** | **0.165** | 0.742 | 0.996 |
| **Recessive model (TT vs. TC+CC)** | | | | | | |
| Asian | 1.32 (1.11, 1.58) | **0.003** | **0.016** | **0.072** | **0.146** | 0.218 |
| Non-Hodgkin lymphoma | 1.48 (1.08, 2.03) | **0.006** | **0.028** | **0.053** | **0.113** | **0.186** |
| HB | 1.15 (1.03, 1.28) | **0.009** | **0.017** | **0.113** | 0.651 | 0.984 |
| Low quality (≤ 9) | 1.16 (1.01, 1.33) | **0.035** | **0.139** | 0.237 | 0.581 | 0.968 |
| **Homozygote model (TT vs. CC)** | | | | | | |
| Overall | 1.08 (1.01, 1.16) | **0.077** | **0.187** | 0.852 | 0.982 | 0.999 |
| Asian | 1.60 (1.31, 1.96) | **0.003** | **0.009** | **0.067** | **0.136** | **0.178** |
| Non-Hodgkin lymphoma | 1.46 (1.04, 2.05) | **0.005** | **0.018** | **0.097** | **0.134** | 0.265 |
| Hepatocellular cancer | 2.24 (1.22, 4.09) | **0.001** | **0.007** | **0.015** | **0.083** | **0.117** |
| HB | 1.30 (1.15, 1.46) | **0.009** | **0.045** | 0.209 | 0.454 | 0.935 |
| Low quality (≤ 9) | 1.23 (1.06, 1.42) | **0.015** | **0.189** | 0.539 | 0.878 | 0.996 |
| **Heterozygote model (TC vs. CC)** | | | | | | |
| Overall | 1.09 (1.04, 1.15) | **0.011** | **0.182** | 0.643 | 0.783 | 0.999 |
| Caucasian | 1.05 (1.00, 1.11) | **0.018** | **0.190** | 0.788 | 0.896 | 0.997 |
| Asian | 1.43 (1.24, 1.66) | **0.007** | **0.013** | **0.145** | 0.411 | 0.832 |
| Bladder cancer | 2.75 (1.48, 5.13) | **0.003** | **0.018** | **0.069** | **0.116** | **0.158** |
| Breast cancer | 1.08 (1.01, 1.16) | **0.025** | **0.146** | 0.367 | 0.686 | 0.997 |
| Hepatocellular cancer | 1.93 (1.18, 3.16) | **0.003** | **0.011** | **0.102** | **0.164** | 0.227 |
| HB | 1.23 (1.13, 1.35) | **0.025** | **0.170** | 0.232 | 0.567 | 0.916 |
| High quality (> 9) | 1.08 (1.02, 1.14) | **0.049** | **0.137** | 0.447 | 0.789 | 0.977 |
| Low quality (≤ 9) | 1.12 (1.01, 1.24) | **0.054** | **0.182** | 0.417 | 0.890 | 0.976 |
| **Allele model (T vs. C)** | | | | | | |
| Overall | 1.05 (1.01, 1.08) | **0.086** | **0.190** | 0.507 | 0.831 | 0.987 |
| Asian | 1.32 (1.20, 1.46) | **0.007** | **0.046** | **0.137** | 0.368 | 0.769 |
| Bladder cancer | 2.92 (1.69, 5.04) | **0.001** | **0.006** | **0.035** | **0.108** | **0.187** |
| Hepatocellular cancer | 1.54 (1.14, 2.07) | **0.006** | **0.034** | **0.117** | **0.181** | 0.370 |
| HB | 1.16 (1.09, 1.23) | **0.011** | **0.064** | **0.116** | **0.192** | 0.473 |
| Low quality (≤ 9) | 1.09 (1.02, 1.17) | **0.043** | **0.158** | 0.438 | 0.799 | 0.986 |

FPRR analysis results were in bold, with a prior probability <0.2. [a] P value for significant test.

## 4. Discussion

It is increasingly evident that genetics is pivotal in influencing cancer risk [1,4]. Given that polymorphism represents a primary source of variation in human genetic information, polymorphisms's association with cancer is gaining much

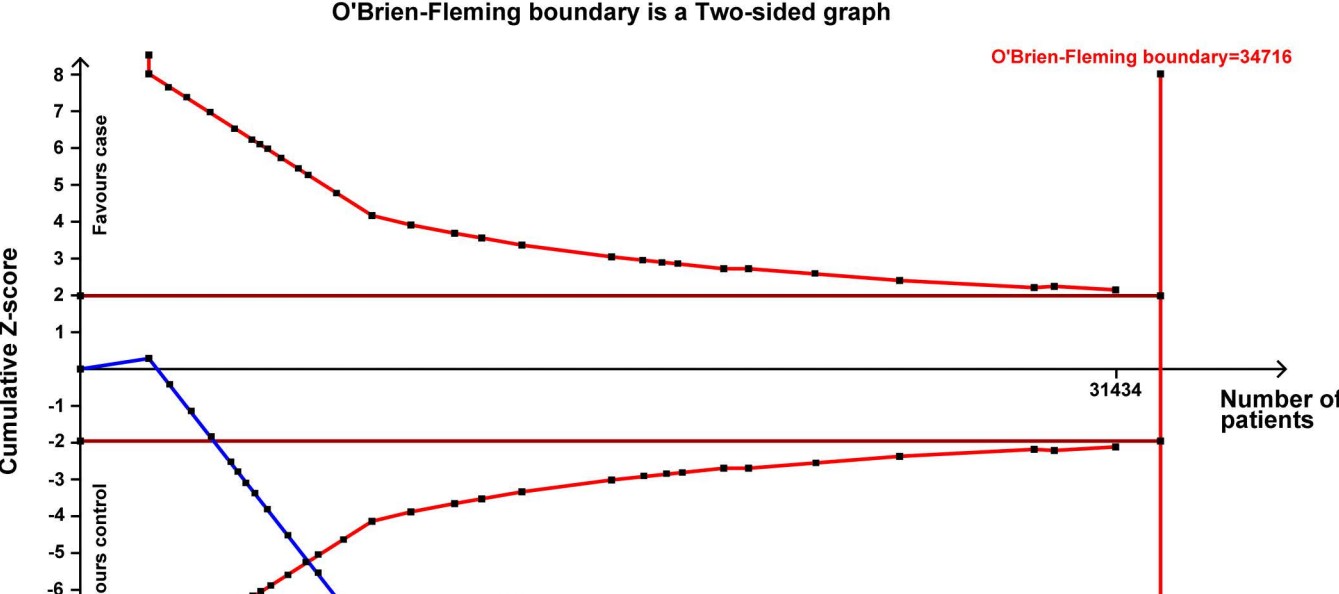

**Fig 6. TSA of rs743572 polymorphism's association (dominant model) with cancer risk.**

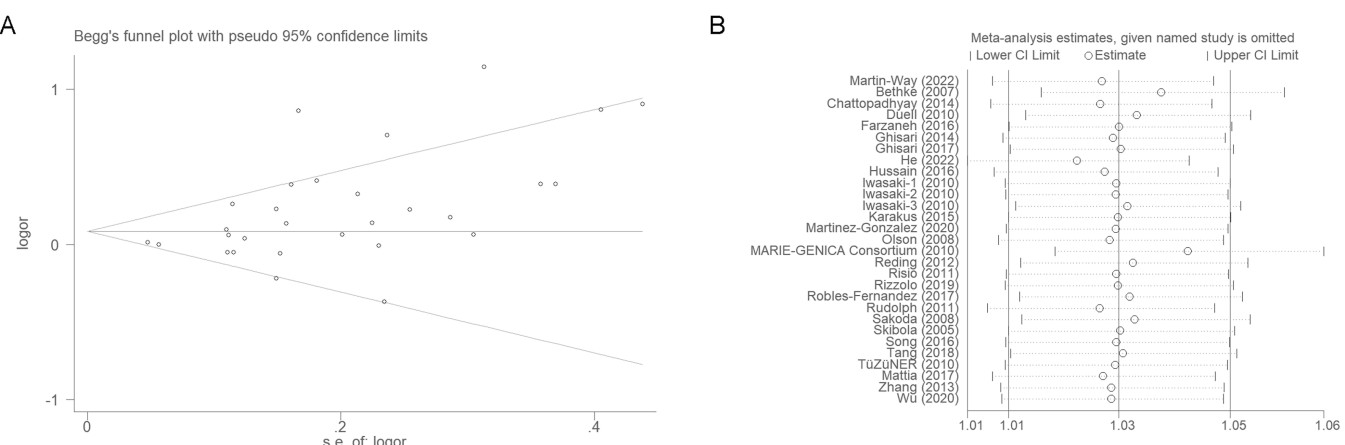

**Fig 7. Publication bias and Sensitivity analysis.**

**Table 5. Begg's and Egger's tests for publication bias.**

| Model | Dominant | Recessive | Homozygote | Heterozygote | Allele |
|---|---|---|---|---|---|
| $P_{Begg}$ | 0.389 | 0.514 | 0.643 | 0.715 | 0.402 |
| $P_{Egger}$ | 0.257 | 0.482 | 0.379 | 0.843 | 0.371 |

attention [3]. Advancements in medical technology have sparked significant interest in tumor genetic susceptibility, leading to a surge in research on genetic polymorphisms related to cancer. Among these, genetic polymorphisms within the cytochrome P450 family have become a pivotal factor in exploring polymorphisms associated with malignant tumor risks [15,19].

CYP17A1 is crucial for intricate cancer biology, particularly in relation to hormone-dependent cancers. CYP17A1 is a pivotal enzyme involved in steroid hormone biosynthesis, including androgens and estrogens [24,34–36]. The dysregulation of these hormones contributes to developing and facilitating diverse cancers, and CYP17A1 emerges as a crucial player in this context. CYP17A1 has been extensively explored for its role in hormone-dependent malignancies, such as prostate cancer and hormone-sensitive breast cancer [19–26]. In our study, we found a significant association between CYP17A1 rs743572 polymorphism with both hormone-sensitive and non-hormone-sensitive cancers, reflecting its dual role in steroid hormone synthesis and broader metabolic pathways. For instance, the increased risk of breast cancer supports its involvement in estrogen biosynthesis, particularly in hormone receptor-positive subtypes. Interestingly, the significant association with bladder cancer may reflect androgen-related mechanisms, as androgen signaling has been implicated in bladder tumorigenesis. These findings highlight the need for further studies stratified by hormone receptor status to clarify cancer type-specific effects.

The enzyme CYP17A1 plays a critical role in steroid hormone biosynthesis by catalyzing two key reactions, 17α-hydroxylase and 17, 20-lyase activities, which convert pregnenolone and progesterone into precursors for glucocorticoids, mineralocorticoids, and sex hormones such as androgens and estrogens [25]. This enzyme is a pivotal target in hormone-dependent cancer therapies, especially in prostate cancer and breast cancer. Prostate cancer, a prevalent malignancy in men, is often driven by androgens, the male sex hormones. Inhibition of CYP17A1 has thus become a targeted strategy to impede androgen synthesis and, consequently, hinder the growth and progression of prostate cancer. Similarly, in hormone-sensitive breast cancer, where estrogen signaling fuels tumor development, CYP17A1 has garnered attention. The enzyme contributes to the synthesis of estrogen precursors, and interventions targeting CYP17A1 have been explored to mitigate estrogen production and dampen the stimulatory effects on breast cancer cells [18,22,27]. This approach is particularly relevant in postmenopausal women, where estrogen produced by the ovaries is shifted to be produced by peripheral tissues, including the adrenal glands, where CYP17A1 is active. However, side effects from glucocorticoid depletion and genetic variability in CYP17A1 can influence treatment efficacy and patient prognosis. Selective CYP17A1 lyase inhibitors offer a promising approach to optimize therapy by reducing side effects while maintaining anti-androgen effects. Abiraterone acetate, a notable CYP17 inhibitor, has shown efficacy in treating metastatic castration-resistant prostate cancer, demonstrating the clinical significance of targeting CYP17A1 in managing advanced stages of the disease [31,33]. While the inhibition of CYP17A1 presents a promising approach in certain cancer types, it is essential to consider potential side effects and the broader impact on steroid hormone balance in the body. Hormonal imbalances induced by CYP17A1 inhibition may lead to adverse effects, underscoring the need for careful clinical management and monitoring.

This review revealed a marked association between rs743572 and elevated cancer susceptibility risk, which was validated through FPRP and TSA analyses. Stratified analyses by cancer type revealed a significant association between rs743572 and bladder cancer, breast cancer, non-Hodgkin lymphoma, and hepatocellular cancer. However, most subgroups had insufficient numbers, which may attenuate the statistical power. Nevertheless, limited data restrain the assessment of rs743572's association with ER-negative and ER-positive breast cancer. Future research with a larger cohort is needed.

This present analysis possesses several strengths: (1) findings were robustly confirmed through TSA analysis, enhancing result reliability; (2) all incorporated studies adhered to the Hardy-Weinberg Equilibrium (HWE), contributing to the overall study reliability; (3) this systematic evaluation represents the inaugural analysis examining the relationship between CYP17A1 rs743572 and cancer risk; (4) FPRP analysis was applied to all significant observations.

Nevertheless, there are certain limitations. (1) The inclusion of subjects was restricted to Caucasians and Asians, lacking representation from other ethnic groups, potentially introducing publication bias. (2) Some subgroups exhibited

a relatively small number of studies on CYP17A1 rs743572, raising the possibility of significant or insignificant outcomes occurring by chance. (3) Detailed information on factors (exposure to radiation and carcinogens, smoking, etc.) was not uniformly available in all included studies, hampering stratification analyses. (4) CYP17A1 plays an essential role in the synthesis of steroid hormones, further studies stratified by hormone receptor status to clarify cancer type-specific effects are needed. Thus, future research should aim for larger sample sizes, incorporate multi-racial populations, and adopt a standardized, multi-center approach to provide more comprehensive and detailed data.

## 5. Conclusions

In summary, this systematic meta-analysis suggests a crucial role for the rs743572 polymorphism in cancer pathogenesis, with particular prominence observed in bladder cancer, breast cancer, non-Hodgkin lymphoma, and hepatocellular cancer—a noteworthy observation supported by FPRP evaluation. However, it is essential to note that future comprehensive studies are warranted to validate and reinforce the robustness of our findings.

## Supporting information

**S1 Table. Score of quality assessment.**
(DOCX)

**S1 File. Original Data.**
(ZIP)

**S2 File. PRISMA_2020_checklist.**
(DOCX)

## Acknowledgments

We would like to acknowledge the reviewers for their helpful comments on this paper.

## Author contributions

**Conceptualization:** Hua Zou.

**Data curation:** Hua Zou.

**Formal analysis:** Hua Zou.

**Investigation:** Hua Zou.

**Methodology:** Hua Zou, Zhumin Cao.

**Project administration:** Zhumin Cao, Bin Wang.

**Software:** Hua Zou.

**Supervision:** Zhumin Cao, Ying Li.

**Visualization:** Hua Zou.

**Writing – original draft:** Hua Zou.

**Writing – review & editing:** Hua Zou, Zhumin Cao, Ying Li, Bin Wang.

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
