## [Decision Letter · Decision Letter 0]

Dear Dr. Wang,

Thank you for submitting your manuscript to PLOS ONE. After careful consideration, we feel that it has merit but does not fully meet PLOS ONE’s publication criteria as it currently stands. Therefore, we invite you to submit a revised version of the manuscript that addresses the points raised during the review process.

**ACADEMIC EDITOR:**

**References: ** Please review and revise the list of references carefully. In particular, **Reference number 3 appears to be incomplete** . Kindly provide the full citation, including the title, journal name, volume, issue, page numbers. Additionally, for **Reference number 34** , please ensure that the **year of publication is mentioned** , along with other complete bibliographic details.**Figures: ****Figures 7A and 7B are not clear**  in the current version of the manuscript. The resolution appears low and important details are difficult to interpret. It is recommended to **replace them with high-resolution images** , ensuring all labels and legends are clearly visible.**Forest Plot: ** A **comment on the forest plot is necessary**  to aid interpretation. Authors are encouraged to include a brief explanation in the results or discussion section about what the forest plot depicts, including the direction and strength of the effect sizes, confidence intervals, and any observed heterogeneity among the studies.

We look forward to receiving your revised manuscript.

Kind regards,

Dr. Charushila Yuvraj Kadam, PhD

Academic Editor

PLOS ONE

Journal Requirements:

2. As required by our policy on Data Availability, please ensure your manuscript or supplementary information includes the following:

4. We noticed you have some minor occurrence of overlapping text with the following previous publication(s), which needs to be addressed:

https://portlandpress.com/bioscirep/article/39/9/BSR20190368/220428/Association-of-the-vitamin-D-metabolism-gene-GC

In your revision ensure you cite all your sources (including your own works), and quote or rephrase any duplicated text outside the methods section. Further consideration is dependent on these concerns being addressed.

This research was funded by grants from Chongqing Young and Middle-aged Medical High-end Talents (No. YXGD202405), Chongqing District and County Head Goose Talents, Chongqing Science and Technology and Health Joint Scientific Research Project on Traditional Chinese Medicine (No. 2024ZYYB036), and Chongqing Banan District Science and Technology and Health Joint Scientific Research Project on Traditional Chinese Medicine (No. BNWJ202300112).

6. In the online submission form, you indicated that the datasets generated during and/or analysed during the current study are available from the corresponding author on reasonable request.

7. Please remove all personal information, ensure that the data shared are in accordance with participant consent, and re-upload a fully anonymized data set.

8. We notice that your supplementary [table] are included in the manuscript file. Please remove them and upload them with the file type 'Supporting Information'. Please ensure that each Supporting Information file has a legend listed in the manuscript after the references list.

Reviewers' comments:

Reviewer's Responses to Questions

**Comments to the Author**

1. Is the manuscript technically sound, and do the data support the conclusions?

Reviewer #1: Yes

Reviewer #2: Yes

2. Has the statistical analysis been performed appropriately and rigorously?

Reviewer #1: I Don't Know

Reviewer #2: No

3. Have the authors made all data underlying the findings in their manuscript fully available?

Reviewer #1: Yes

Reviewer #2: Yes

4. Is the manuscript presented in an intelligible fashion and written in standard English?

Reviewer #1: Yes

Reviewer #2: Yes

Reviewer #1: The meta analysis covers an important topic for cancer etiology, although few papers have issued this point but the meta analysis covered most of the publications. The methodology and materials are accepted, the figures and tables are accepted, discussion and results are accepted

Reviewer #2: The authors have conducted an important meta analysis on the association between CYP17A1 rs743572 polymorphism and cancer susceptibility. They have presented a well written manuscript and have presented their findings appropriately. However, to improve on the manuscript the authors should address the following comments.

1. Considering that CYP17A1 is involved in the synthesis of steroid hormones, the findings should show the significance of the enzyme by comparing its impact on hormone sensitive and non-hormone sensitive cancers.

2. The effect of the enzyme on treatment using steroid and hormone therapy and how this could affect disease progression and outcome in patients.

**Do you want your identity to be public for this peer review?** For information about this choice, including consent withdrawal, please see our Privacy Policy

Reviewer #1: **Yes: ** Haytham Abdelkader

Reviewer #2: No

---

## [Author Response · Author response to Decision Letter 1]

29 May 2025

ACADEMIC EDITOR:

1.References: Please review and revise the list of references carefully. In particular, Reference number 3 appears to be incomplete. Kindly provide the full citation, including the title, journal name, volume, issue, page numbers. Additionally, for Reference number 34, please ensure that the year of publication is mentioned, along with other complete bibliographic details.

Response: Thanks for your kind help. We have rectified the reference format as you suggested, please check MS for details.

2. Figures: Figures 7A and 7B are not clear in the current version of the manuscript. The resolution appears low and important details are difficult to interpret. It is recommended to replace them with high-resolution images, ensuring all labels and legends are clearly visible.

Response: We have provided higher-resolution versions of Figures 7A and 7B with clearer labels and legends in the revised manuscript. Please check them for details. Thanks very much for your helpful suggestions.

3. Forest Plot: A comment on the forest plot is necessary to aid interpretation. Authors are encouraged to include a brief explanation in the results or discussion section about what the forest plot depicts, including the direction and strength of the effect sizes, confidence intervals, and any observed heterogeneity among the studies.

 Response: Thanks for these valuable comments. We have added the detailed information about the forest plots in the Figure legends. Please check Figures 2-5 for details.

A rebuttal letter that responds to each point raised by the academic editor and reviewer(s). You should upload this letter as a separate file labeled 'Response to Reviewers'.

A marked-up copy of your manuscript that highlights changes made to the original version. You should upload this as a separate file labeled 'Revised Manuscript with Track Changes'.

An unmarked version of your revised paper without tracked changes. You should upload this as a separate file labeled 'Manuscript'.

We look forward to receiving your revised manuscript.

Kind regards,

Dr. Charushila Yuvraj Kadam, PhD

Academic Editor

PLOS ONE

Journal Requirements:

 Response: Confirmed.  

2. As required by our policy on Data Availability, please ensure your manuscript or supplementary information includes the following:

Response: Thanks very much for the valuable information. We have carefully checked each item to ensure data availability; all the requirements you mentioned above are included in the current MS.

 Response: Thanks. We have registered and validated the corresponding author's ORCID iD in Editorial Manager as required. The information has been successfully updated in the submission system.

 4. We noticed you have some minor occurrence of overlapping text with the following previous publication(s), which needs to be addressed:

https://portlandpress.com/bioscirep/article/39/9/BSR20190368/220428/Association-of-the-vitamin-D-metabolism-gene-GC

In your revision ensure you cite all your sources (including your own works), and quote or rephrase any duplicated text outside the methods section. Further consideration is dependent on these concerns being addressed.

Response: Thank you for bringing this to our attention. We will properly cite the reference and rephrase any overlapping text in the revised manuscript.

This research was funded by grants from Chongqing Young and Middle-aged Medical High-end Talents (No. YXGD202405), Chongqing District and County Head Goose Talents, Chongqing Science and Technology and Health Joint Scientific Research Project on Traditional Chinese Medicine (No. 2024ZYYB036), and Chongqing Banan District Science and Technology and Health Joint Scientific Research Project on Traditional Chinese Medicine (No. BNWJ202300112).

Response: Thanks for your kind help. We have added the detailed information about the role of funders in the MS, please check “Financial disclosure” for details.

6. In the online submission form, you indicated that the datasets generated during and/or analysed during the current study are available from the corresponding author on reasonable request.

 Response: Confirmed.

7. Please remove all personal information, ensure that the data shared are in accordance with participant consent, and re-upload a fully anonymized data set.

Response: Confirmed.  

8. We notice that your supplementary [table] are included in the manuscript file. Please remove them and upload them with the file type 'Supporting Information'. Please ensure that each Supporting Information file has a legend listed in the manuscript after the references list.

Response: Thank you for your kind advice. We have now removed the supplementary tables from the main manuscript and uploaded them as separate Supporting Information files, with corresponding legends included after the reference list. Please check it for details.

 Response: Confirmed.

 Response: Confirmed.

Reviewers' comments:

Reviewer's Responses to Questions

Comments to the Author

1. Is the manuscript technically sound, and do the data support the conclusions?

Reviewer #1: Yes

Reviewer #2: Yes

2. Has the statistical analysis been performed appropriately and rigorously?

Reviewer #1: I Don't Know

Reviewer #2: No

3. Have the authors made all data underlying the findings in their manuscript fully available?

Reviewer #1: Yes

Reviewer #2: Yes

4. Is the manuscript presented in an intelligible fashion and written in standard English?

Reviewer #1: Yes

Reviewer #2: Yes

5. Review Comments to the Author

Reviewer #1: The meta analysis covers an important topic for cancer etiology, although few papers have issued this point but the meta analysis covered most of the publications. The methodology and materials are accepted, the figures and tables are accepted, discussion and results are accepted

Response: Thanks very much for your positive comments.

Reviewer #2: The authors have conducted an important meta analysis on the association between CYP17A1 rs743572 polymorphism and cancer susceptibility. They have presented a well written manuscript and have presented their findings appropriately. However, to improve on the manuscript the authors should address the following comments.

1. Considering that CYP17A1 is involved in the synthesis of steroid hormones, the findings should show the significance of the enzyme by comparing its impact on hormone sensitive and non-hormone sensitive cancers.

Response: We sincerely appreciate your valuable comments about the importance of differentiating the impact of the CYP17A1 rs743572 polymorphism on hormone-sensitive and non-hormone-sensitive cancers. Our meta-analysis revealed the potential associations with both cancer types, consistent with CYP17A1's dual roles in steroid hormone synthesis and broader metabolic pathways. For instance, in breast cancer, the observed risk elevation (dominant model OR=1.08, 95% CI=1.01–1.15) aligns with CYP17A1’s function in estrogen precursor synthesis, particularly in hormone receptor-positive breast cancer. Interestingly, we also noticed the unexpected associations with bladder cancer (dominant, OR=3.14, 95% CI=1.70-5.80) may reflect underlying mechanisms (androgen signaling). We acknowledge that these results are relatively preliminary, further stratification analysis by hormone receptor status is still needed for a deeper clarification. We have added the relevant contents in the Discussion section of the latest MS, please check it for more information: “In our study, we found a significant association between CYP17A1 rs743572 polymorphism with both hormone-sensitive and non-hormone-sensitive cancers, reflecting its dual role in steroid hormone synthesis and broader metabolic pathways. For instance, the increased risk of breast cancer supports its involvement in estrogen biosynthesis, particularly in hormone receptor-positive subtypes. Interestingly, the significant association with bladder cancer may reflect androgen-related mechanisms, as androgen signaling has been implicated in bladder tumorigenesis. These findings highlight the need for further studies stratified by hormone receptor status to clarify cancer type-specific effects”; “(4) CYP17A1 plays an essent

---

## [Editor Report · Decision Letter 1]

Association between CYP17A1 rs743572 polymorphism and cancer risk: a meta-analysis

PONE-D-24-52835R1

Dear Dr. Bin Wang,

We’re pleased to inform you that your manuscript has been judged scientifically suitable for publication and will be formally accepted for publication once it meets all outstanding technical requirements.

Kind regards,

Charushila Yuvraj Kadam, PhD

Academic Editor

PLOS ONE

---

## [Editor Report · Acceptance letter]

PONE-D-24-52835R1

PLOS ONE

Dear Dr. Wang,

I'm pleased to inform you that your manuscript has been deemed suitable for publication in PLOS ONE. Congratulations! Your manuscript is now being handed over to our production team.

Kind regards,

on behalf of

Dr. Charushila Yuvraj Kadam

Academic Editor

PLOS ONE